# E-CRF: Embedded Conditional Random Field for Boundary-caused Class Weights Confusion in Semantic Segmentation

**Jie Zhu**[1,2]    **Huabin Huang**[3]    **Banghuai Li**[3]    **Leye Wang**[1,2*]

Key Lab of High Confidence Software Technologies (Peking University), Ministry of Education, China[1]
School of Computer Science, Peking University, Beijing, China[2]
MEGVII Technology[3]
zhujie@stu.pku.edu.cn, {huanghuabin1994, libanghuai}@gmail.com, leyewang@pku.edu.cn

## Abstract

Modern semantic segmentation methods devote much effect to adjusting image feature representations to improve the segmentation performance in various ways, such as architecture design, attention mechnism, etc. However, almost all those methods neglect the particularity of class weights (in the classification layer) in segmentation models. In this paper, we notice that the class weights of categories that tend to share many adjacent boundary pixels lack discrimination, thereby limiting the performance. We call this issue *Boundary-caused Class Weights Confusion* (**BCWC**). We try to focus on this problem and propose a novel method named *Embedded Conditional Random Field* (**E-CRF**) to alleviate it. E-CRF innovatively fuses the CRF into the CNN network as an organic whole for more effective end-to-end optimization. The reasons are two folds. It utilizes CRF to guide the message passing between pixels in high-level features to purify the feature representation of boundary pixels, with the help of inner pixels belonging to the same object. More importantly, it enables optimizing class weights from both **scale** and **direction** during backpropagation. We make detailed theoretical analysis to prove it. Besides, superpixel is integrated into E-CRF and served as an auxiliary to exploit the local object prior for more reliable message passing. Finally, our proposed method yields impressive results on ADE20K, Cityscapes, and Pascal Context datasets.

## 1 Introduction

Semantic segmentation plays an important role in practical applications such as autonomous driving, image editing, *etc*. Nowadays, numerous CNN-based methods (Chen et al., 2014; Fu et al., 2019; Ding et al., 2019) have been proposed. They attempt to adjust the image feature representation of the model itself to recognize each pixel correctly. However, almost all those methods neglect the particularity of class weights (in the classification layer) that play an important role in distinguishing pixel categories in segmentation models. Hence, it is critical to keep class weights discriminative. Unfortunately, CNN models have the natural defect for this. Generally speaking, most discriminative higher layers in the CNN network always have the larger receptive field, thus pixels around the boundary may obtain confusing features from both sides. As a result, these ambiguous boundary pixels will mislead the optimization direction of the model and make the class weights of such categories that tend to share adjacent pixels indistinguishable. For the convenience of illustration, we call this issue as *Boundary-caused Class Weights Confusion* (**BCWC**). We take DeeplabV3+ (Chen et al., 2018a) as an example to train on ADE20K (Zhou et al., 2017) dataset. Then, we count the number of adjacent pixels for each class pair and find a corresponding category that has the most adjacent pixels for each class. Fig 1(a) shows the similarity of the class weight between these pairs in descending order according to the number of adjacent pixels. It is clear that if two categories share more adjacent pixels, their class weights tend to be more similar, which actually indicates that BCWC makes class representations lack discrimination and damages the overall segmentation performance.

---

*Corresponding author.

Previous works mainly aim to improve boundary pixel segmentation, but they seldom explicitly take class weights confusion *i.e.*, BCWC, into consideration [1].

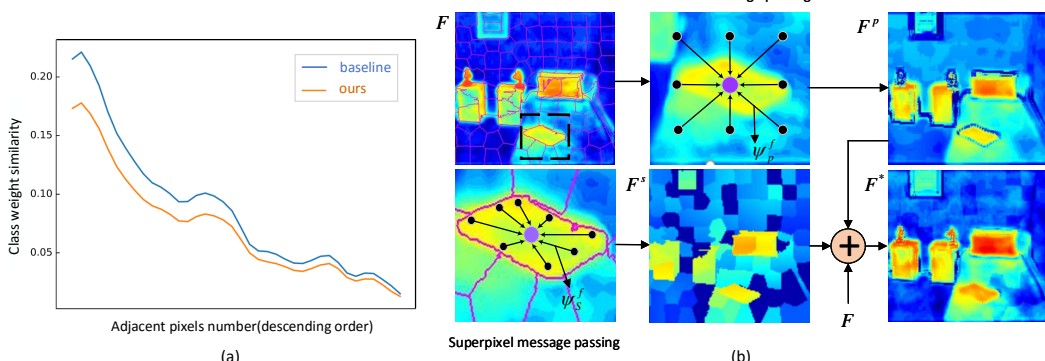

Figure 1: **(a)** Observations on ADE20K. We find a corresponding category that shares the most adjacent pixels for each class and calculate the similarity of their class weights. X-axis stands for the number of adjacent pixels for each class pair in descending order, and Y-axis represents the similarity of their class weights. Blue line denotes baseline model while orange line denotes E-CRF. Apparently, two categories that share more adjacent pixels are inclined to have more similar class weights, while E-CRF effectively decreases the similarity between *adjacent categories* and makes their class weights more discriminative. **(b)** Message passing procedure of E-CRF. $F$ is the original feature maps of the CNN network. E-CRF utilizes pairwise module $\psi_p^f$ and auxiliary superpixel-based module $\psi_s^f$ on $F$ to obtain refined feature maps $F^p$ and $F^s$ respectively. Then $F$, $F^p$ and $F^s$ are fused as $F^*$ to further segment the image.

Considering the inherent drawback of CNN networks mentioned before, delving into the relationship between raw pixels becomes a potential alternative to eliminate the BCWC problem, and Conditional Random Field (**CRF**) (Chen et al., 2014) stands out. It is generally known that pixels of the same object tend to share similar characteristics in the local area. Intuitively, CRF utilizes the local consistency between original image pixels to refine the boundary segmentation results with the help of inner pixels of the same object. CRF makes some boundary pixels that are misclassified by the CNN network quite easy to be recognized correctly. But these CRF-based methods (Chen et al., 2014; Zhen et al., 2020a) only adopt CRF as an offline post-processing module, we call it *Vanilla-CRF*, to refine the final segmentation results. They are incapable of relieving BCWC problem as CRF and the CNN network are treated as two totally separate modules.

Based on Chen et al. (2014; 2017a), Lin et al. (2015); Arnab et al. (2016); Zheng et al. (2015) go a step further to unify the segmentation model and CRF in a single pipeline for end-to-end training. We call it *Joint-CRF* for simplicity. Same as *Vanilla-CRF*, *Joint-CRF* inclines to rectify those misclassified boundary pixels via increasing the prediction score of the associated category, which means it still operates on the *object class probabilities*. But it can alleviate the BCWC problem to some extent as the probability score refined by CRF directly involves in the model backpropagation. Afterwards, the disturbing gradients caused by those pixels will be relieved, which will promote the class representation learning. However, as shown in Fig 3, the effectiveness of Joint-CRF is restricted as it only optimizes the scale of the gradient and lacks the ability to optimize class representations effectively due to the defective design. More theoretical analysis can be found in Sec. 3.3.

To overcome the aforementioned drawbacks, in this paper, we present a novel approach named *Embedded CRF* (**E-CRF**) to address the BCWC problem more effectively. The superiority of E-CRF lies in two main aspects. On the one hand, by fusing CRF mechanism into the segmentation model, E-CRF utilizes the local consistency among original image pixels to guide the message passing of high-level features. Each pixel pair that comes from the same object tends to obtain higher message passing weights. Therefore, the feature representation of the boundary pixels can be purified by the corresponding inner pixels from the same object. In turn, those pixels will further contribute to the discriminative class representation learning. On the other hand, it extends the fashion of optimizing class weights from one perspective (*i.e.*, scale) to two (*i.e.*, scale and direction) during

---

[1]These methods improve boundary segemetation and may have effect on class weights. But they are not explicit and lack theoretical analysis. We show great benifit of explicitly considering BCWC issue. See A.4.1.

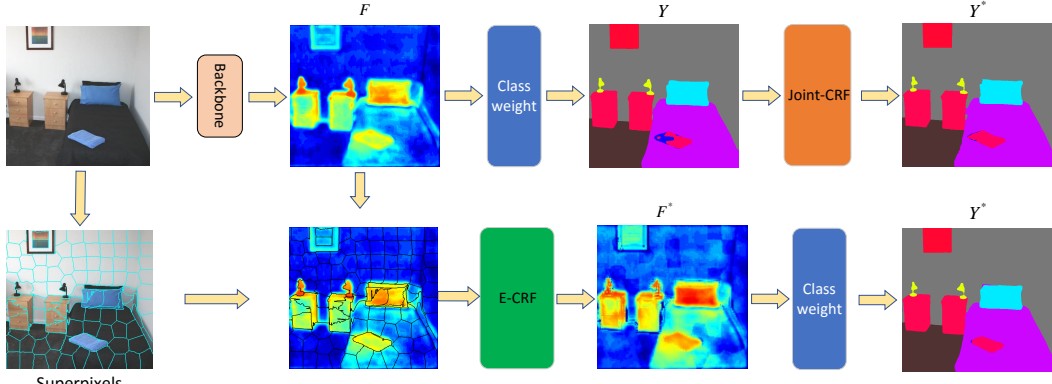

Figure 2: Illustration of Joint-CRF and E-CRF. The first row is the simplified structure of Joint-CRF, which unifies the CNN network and CRF in a single pipeline for end-to-end training. However, CRF only serves as a post-processing module. The second row is the overview of our E-CRF, which fuses CRF into CNN network as an organic whole to eliminate BCWC problem.

backpropagation. In Sec. 3.3, we prove theoretically that E-CRF outperforms other CRF-based methods on eliminating the BCWC problem by optimizing both direction and scale of the disturbing gradient of class weights. However, during this process, the noise information can also have a direct influence on the class weights (likely to hinder the optimization for the BCWC problem). In addition, E-CRF adopts superpixel (Ren & Malik, 2003) as an auxiliary and leverage its local prior to suppress the noise and further strengthen the reliability of the message passing to the boundary pixels. Superpixel groups adjacent pixels that share similar characteristics to form a block. It is prone to achieve clear and smooth boundaries and increases the potential for higher segmentation performance. In E-CRF, we average the deep feature representation of all inner pixels in the same superpixel block and then add this local object prior to each pixel back to enhance the representation of boundary pixels.

In this work, we explicitly propose the BCWC problem in semantic segmentation and an effective approach to alleviate it. We conduct extensive experiments on three challenging semantic segmentation benchmarks, *i.e.*, ADE20K (Zhou et al., 2017), Cityscapes (Cordts et al., 2016), and Pascal Context (Mottaghi et al., 2014), and yeild impressive results. For example, E-CRF outperforms baselines (DeeplabV3+ (Chen et al., 2018a) with ResNet-101 (He et al., 2016)) by **1.42**% mIoU on ADE20K and **0.92**% mIoU on Cityscapes in single scale. In addition, we make an exhaustive theoretical analysis in Sec. 3.3 to prove the effectiveness of E-CRF. Code is available at `https://github.com/JiePKU/E-CRF`.

## 2 RELATED WORK

**Semantic Segmentation.** Fully convolutional network (FCN) (Long et al., 2015) based methods have made great progress in semantic segmentation by leveraging the powerful convolutional features of classification networks (He et al., 2016; Huang et al., 2017) pre-trained on large-scale data (Russakovsky et al., 2015). There are several model variants proposed to enhance contextual aggregation. For example, DeeplabV2 (Chen et al., 2017a) and DeeplabV3 (Chen et al., 2017b) take advantage of the astrous spatial pyramid pooling (ASPP) to embed contextual information, which consists of parallel dilated convolutions with different dilated rates to broaden the receptive field. Inspired by the encoder-decoder structures (Ronneberger et al., 2015; Ding et al., 2018), DeeplabV3+ (Chen et al., 2018a) adds a decoder upon DeeplabV3 to refine the segmentation results especially along object boundaries. With the success of self-attention mechanism in natural language processing, Non-local (Wang et al., 2018) first adopts self-attention mechanism as a module for computer vision tasks, such as video classification, object detection and instance segmentation. $A^2$Net (Chen et al., 2018b) proposes the double attention block to distribute and gather informative global features from the entire spatio-temporal space of the images.

**Conditional Random Fields.** Fully connected CRFs have been used for semantic image labeling in (Payet & Todorovic, 2010; Toyoda & Hasegawa, 2008), but inference complexity in fully connected models has restricted their application to sets of hundreds of image regions or fewer. To address this issue, densely connected pairwise potentials (Krähenbühl & Koltun, 2011) facilitate interactions

between all pairs of image pixels based on a mean field approximation to the CRF distribution. Chen et al. (2014) show further improvements by post-processing the results of a CNN with a CRF. Subsequent works (Lin et al., 2015; Arnab et al., 2016; Zheng et al., 2015) have taken this idea further by incorporating a CRF as layers within a deep network and then learning parameters of both the CRF and CNN together via backpropagation. In terms of enhancements to conventional CRF models, Ladický et al. (2010) propose using an off-the-shelf object detector to provide additional cues for semantic segmentation.

**Superpixel.** Superpixel (Ren & Malik, 2003) is pixels with similar characteristics that are grouped together to form a large block. Since its introduction in 2003, there have been many mature algorithms (Achanta et al., 2012; Weikersdorfer et al., 2013; Van den Bergh et al., 2012). Owing to their representational and computational efficiency, superpixels are widely-used in computer vision algorithms such as target detection (Shu et al., 2013; Yan et al., 2015), semantic segmentation (Gould et al., 2008; Sharma et al., 2014; Gadde et al., 2016), and saliency estimation (He et al., 2015; Perazzi et al., 2012). Yan et al. (2015) convert object detection problem into superpixel labeling problem and conducts an energy function considering appearance, spatial context and numbers of labels. Gadde et al. (2016) use superpixels to change how information is stored in the higher level of a CNN. In (He et al., 2015), superpixels are taken as input and contextual information is recovered among superpixels, which enables large context to be involved in analysis.

We give a detailed discussion about the difference between E-CRF and three highly related works including PCGrad (Yu et al., 2020b), OCNet (Yuan & Wang, 2018), and SegFix (Yuan et al., 2020b) in Appendix A.5.

## 3 METHOD

### 3.1 REVISITING CONDITIONAL RANDOM FIELD (CRF)

CRF is a typical discriminative model suitable for prediction tasks where contextual information or the state of the neighbors affects the current prediction. Nowadays, it is widely adopted in the semantic segmentation field (Krähenbühl & Koltun, 2011; Chen et al., 2014). CRF utilizes the correlation between original image pixels to refine the segmentation results by modeling this problem as the maximum a posteriori (MAP) inference in a conditional random field (CRF), defined over original image pixels. In practice, the most common way is to approximate CRF as a message passing procedure among pixels and it can be formulated as:

$$Y_i^* = \frac{1}{Z_i} \left( \psi_u(i) + \sum_{j \neq i}^{G} \psi_p(i, j) Y_j \right), \tag{1}$$

where $Y_i$ and $Y_i^*$ are defined as the classification scores of CNN model and CRF respectively for pixel $i$, $Z_i$ is the normalization factor known as the partition function, and $\psi_u(i)$ is a unary function which often adopts $Y_i$ as the default value. $G$ is the associated pixel set with pixel $i$. For example, DenseCRF (Krähenbühl & Koltun, 2011) takes all other pixels except pixel $i$ itself as the set $G$. Moreover, the pairwise function $\psi_p(i, j)$ is defined to measure the message passing weight from pixel $j$ to pixel $i$. It is formulated as:

$$\psi_p(i, j) = \mu(i, j) \underbrace{\sum_{m=1}^{M} \omega^{(m)} k^{(m)}(\mathbf{f}_i, \mathbf{f}_j)}_{k(\mathbf{f}_i, \mathbf{f}_j)}, \tag{2}$$

where $\mu(i, j)$ is a label compatibility function that introduces the co-occurrent probability for a specific label pair assignment at pixel $i$ and $j$, while $k(\mathbf{f}_i, \mathbf{f}_j)$ is a set of hand-designed Gaussian kernels, $\mathbf{f}_i$ and $\mathbf{f}_j$ are feature vectors of pixel $i$ and $j$ in any arbitrary feature space, such as RGB images. $w^{(m)}$ is the corresponding linear combination weight for each Gaussian kernel. When dealing with multi-class image segmentation, $M$=2 is a common setting. Then, $k(\mathbf{f}_i, \mathbf{f}_j)$ is carefully designed as contrast-sensitive two-kernel potentials, defined in terms of color vectors ($I_i$, $I_j$) and

position coordinates $(p_i, p_j)$ for pixel $i$ and $j$ respectively:

$$k(\mathbf{f}_i, \mathbf{f}_j) = w^{(1)} \underbrace{\exp\left(-\frac{|p_i - p_j|^2}{2\theta_\alpha^2} - \frac{|I_i - I_j|^2}{2\theta_\beta^2}\right)}_{appearance \quad kernel} + w^{(2)} \underbrace{\exp\left(-\frac{|p_i - p_j|^2}{2\theta_\gamma^2}\right)}_{smoothness \quad kernel}. \tag{3}$$

The appearance kernel is inspired by the observation that nearby pixels with similar colors are more likely to share the same class. $\theta_\alpha$ and $\theta_\beta$ are scale factors to control the degree of these two elements, *i.e.*, similarity and distance between two pixels. Apart from this, the smoothness kernel further removes the influence of some small isolated regions (Krähenbühl & Koltun, 2011) and $\theta_\gamma$ is the associated scale factor. Notably, all these parameters are learnable during the model training.

Unfortunately, current CRF-based methods (Chen et al., 2014; 2017a; Lin et al., 2015; Liu et al., 2015) for semantic segmentation always adopt CRF as a post-processing module. For example, Vanilla-CRF (Chen et al., 2014; 2017a) utilizes CRF to refine segmentation scores offline, which has no impacts on BCWC since the CNN network and CRF are treated as two separate modules. Joint-CRF (Lin et al., 2015; Liu et al., 2015; Lin et al., 2016) works in a similar way although CRF involves in the backpropagation of CNN networks, restricting its ability to relieve BCWC.

## 3.2 EMBEDDED CRF

To solve the BCWC problem in a more intrinsical way, we propose a novel method named *Embedded CRF* (E-CRF) to tackle the tough problem via fusing the CRF mechanism into the CNN network as an organic whole for more effective end-to-end training. An overview of E-CRF can be found in Fig 2 and we formulate its core function based on Eq (1) as:

$$F_i^* = \frac{1}{Z_i} \left\{ \psi_u^f(i) + \sum_{j \neq i}^G \psi_p^f(i,j) F_j + F_i^S \right\}. \tag{4}$$

Specifically, the first two terms are analogous to Eq (1) but we perform CRF mechanism on the high-level features. $F_i$ stands for the original output of feature extractors for pixel $i$, $\psi_u^f(i)$ and $\psi_p^f(i,j)$ play the same role as they do in Eq (1). $\psi_u^f(i)$ takes $F_i$ as the default value. In addition, we reformulate $\psi_p^f(i,j)$ to perform message passing between pixel pairs in the high-level feature:

$$\psi_p^f(i,j) = \mu^f(i,j) k(\mathbf{f}_i, \mathbf{f}_j). \tag{5}$$

It is worth noting that $k(\mathbf{f}_i, \mathbf{f}_j)$ is no longer hand-designed Gaussian kernels as it is in Eq (1) but simple convolution operators instead to make the whole model more flexible for end-to-end training and optimization. Experiments in Sec. 4 prove this modification is a more suitable choice:

$$k(\mathbf{f}_i, \mathbf{f}_j) = \mathbf{f}_i \cdot \mathbf{f}_j = conv([I_i, p_i]) \cdot conv([I_j, p_j]), \tag{6}$$

where $[x, y]$ denotes the concatenation operator. Different from Eq (3), we normalize the input image $I$ into the range $[0, 1]$ to eliminate the scale variance between pixels and we replace original absolute position coordinates $p$ with *cosine* position embeddings (Vaswani et al., 2017) to make it more compatible with CNN networks. E-CRF encodes the appearance and position of pixels into more discriminative tokens via the flexible convolution operation, then the dot product is adopted to measure the similarity between pixel pairs. As indicated in Eq (6), E-CRF intends to make nearby pixel pairs that share same appearance to achieve higher $k(\mathbf{f}_i, \mathbf{f}_j)$. Its intention is the same as Eq (3). Correspondingly, we also adjust $\mu^f(i,j)$ as the feature compatibility to measure the co-occurrent probability of $F_i$ and $F_j$:

$$\mu^f(i,j) = sigmoid(conv[F_i, F_j]). \tag{7}$$

Another component in Eq (4) is $\boldsymbol{F_i^S}$. It relies on the superpixel algorithm (Ren & Malik, 2003; Weikersdorfer et al., 2013; Van den Bergh et al., 2012; Gadde et al., 2016) to divide the whole image $I$ into several non-overlapping blocks. Pixels in the same superpixel block tend to share the same characteristics. Thus, we adopt this local object prior to achieve the more effective message passing between pixels in the high-level feature space. Concretely, we design $F_i^S$ as:

$$F_i^S = \sum_l^Q \psi_s^f(l) F_l = \sum_l^Q \frac{1}{n} F_l \tag{8}$$

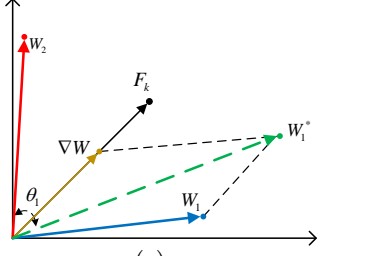 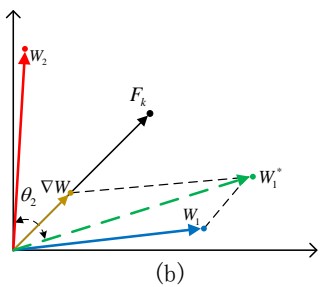 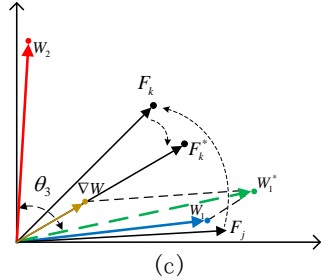

|  (a) | (b) | (c) |

Figure 3: Different optimization effects for baseline, Joint-CRF and E-CRF. $W_1$ and $W_2$ are two class weight vectors that share adjacent pixels. $\nabla W$ is the gradient variation for $W_1$ and $W_1^*$ is the new class weight after gradient descent. $F_k$ is a sample boundary pixel whose ground-truth label keeps consistent with $W_1$ but contains confusing features from both sides. $\theta$ measures the distance between $W_2$ and $W_1^*$. (a) $\nabla W$ tends to push $W_1$ towards $W_2$ due to the confusing features from both classes. (b) Joint-CRF eases the disturbing gradients and reduces the scale of $\nabla W$. Obviously, $\theta_2$ is larger than $\theta_1$. (c) E-CRF aims to enhance the feature representation of $F_k$ via the inner pixels like $F_j$ from the same object. It adjusts both scale and direction of $\nabla W$ to make $\theta_3 > \theta_2 > \theta_1$.

$Q$ is the associated superpixel block that contains pixel $i$ and $\psi_s^f(l)$ devotes the re-weighting factor for the deep features of pixel $l$ in $Q$. We adopt $\psi_s^f(l) = \frac{1}{n}$ and $n$ is the total number of pixels in $Q$. $F_i^S$ serves as a supplement in Eq (4) to add the local object prior to each pixel back, which increases the reliability of message passing in E-CRF. What's more, superpixel (Gould et al., 2008; Sharma et al., 2014; Gadde et al., 2016) always tends to generate clearer and smoother boundary segmentation results than traditional CNN networks or CRFs do, which also increases the potential for more accurate segmentation results. Detailed experiments can be found in Sec. 4.

### 3.3 How E-CRF Relieves BCWC

In this section, without loss of generality, we take multi-class segmentation problem as an example to dive into the principle of E-CRF from the perspective of gradient descent. Suppose $F_k$ is the feature vector of a foreground boundary pixel $k$ whose class label is $c \in [0, n-1]$ and its prediction probability is $P_k^c$. Then, considering the label $y_k$ is one-hot form, the typical cross-entropy loss $L_k$ can be defined as:

$$L_k = -\sum_{i=0}^{i<n} y_k^i \ln P_k^i = -\ln P_k^c, \tag{9}$$

$$P_k^c = softmax(Y_k^c) = \frac{e^{Y_k^c}}{(\sum_{m \neq c} e^{Y_k^m}) + e^{Y_k^c}}, \quad and \quad Y_k^c = W_c^T \cdot F_k, \tag{10}$$

where $W_c$ is the class weight of $c$-th category. $Y_k^m$ is calculated by other class weights and unrelated with $W_c$ and $Y_k^c$. Below the gradient variation $\nabla W_c$ can be formulated as:

$$\nabla W_c = \frac{\partial L_k}{\partial W_c} = \frac{\partial L_k}{\partial P_k^c} \cdot \frac{\partial P_k^c}{\partial Y_k^c} \cdot \frac{\partial Y_k^c}{\partial W_c} \tag{11}$$

Through Eq (9), Eq (10) and Eq (11), class weight in the next iteration will be updated [2]:

$$W_c^* = W_c - \nabla W_c = W_c + (1 - P_k^c) \cdot F_k \tag{12}$$

As shown in Eq (12), the direction of the gradient descent keeps the same as $F_k$ while the magnitude of the gradient is decided by $P_k^c$. *What happens if we integrate the CRF into the segmentation pipeline?* As we have discussed in Sec. 1, Vanilla-CRF has nothing to do with the optimization process of CNN networks, while if we adopt Joint-CRF, $\nabla W_c$ can be reformulated as:

$$-\nabla W_c = (1 - \hat{P}_k^c) \cdot F_k = \underbrace{(1 - \frac{1}{Z_k}(\sum_{j \in G} w_j P_j^c + P_k^c))}_{scale} \cdot F_k \tag{13}$$

---

[2]The detailed derivation process can be found in our *Appendix A.1*.

where $\hat{P}_k^c$ is the refined score by CRF, $w_j$ is the message passing weight from pixel $j$ to pixel $k$ and $P_j^c$ is the original score of pixel $j$. In general, boundary pixel $k$ is hard to classify correctly due to the confusing features from both sides. Thus the original probability $P_k^c$ is always small. In contrast, other inner pixels of the same object are easy to recognize and tend to achieve a higher probability. Consequently, $\hat{P}_k^c$ is usually larger than $P_k^c$ and disturbing gradients caused by boundary pixel will be relieved to some extent, which makes inter-class distance further as shown in Fig 3(b). However, Eq (13) only adjusts the scale of the gradient descent while the direction still keeps the same as $F_k$, which weakens its effects for better representation learning. When it comes to our proposed E-CRF, $\nabla W_c$ can be further defined as:

$$-\nabla W_c = (1 - P_k^{c*}) \cdot F_k^* = \underbrace{(1 - P_k^{c*})}_{scale} \cdot \underbrace{\frac{1}{Z_k}(\sum_{j \in G} w_j F_j + F_k)}_{direction} \tag{14}$$

$$P_k^{c*} = softmax(\sum_{j \in G} w_j Y_j^c + Y_k^c) \tag{15}$$

where $F_k^*$ is the refined feature representations by E-CRF, and $P_k^{c*}$ is the refined score which is analogous to $\hat{P}_k^c$ in Eq (13). Comparing with Joint-CRF, it is clear that E-CRF not only changes the scale of the gradient descent but also adjusts its optimization direction. The optimization process is directly applied to the class weight matrix (in the final layer), which opens up room for more discriminative class weights. In other words, we can adjust the class weight from both the scale and direction to make the class weights more discriminative to decrease the class weights similarity (or class weights confusion). As depicted in Fig 3(c), assume $W_1$ is the class weight vector that a pixel belongs to, while $W_2$ is the other one which has a higher co-occurrent probability with $W_1$ in the same image. E-CRF designs an effective message passing procedure to purify the feature representation of boundary pixels assisted by inner pixels from the same object ($F_j$ in Fig 3(c)). In this way, it relieves the influence of disturbing gradients and makes the inter-class distance between $W_1(W_1^*)$ and $W_2$ further, which means more discriminative feature representations.

## 4 EXPERIMENT

### 4.1 IMPLEMENTATION DETAILS

We follow the previous works (Chen et al., 2014; He et al., 2019b; Chen et al., 2018a) and perform experiments on three challenging semantic segmentation benchmarks, *i.e.*, ADE20K (Zhou et al., 2017), Cityscapes (Cordts et al., 2016) and Pascal Context (Mottaghi et al., 2014). Due to the space limit, a detailed description of these three datasets can be found in our *Appendix A*. We adopt DeeplabV3+ (Chen et al., 2018a) with ResNet (He et al., 2016) pretrained on ImageNet (Russakovsky et al., 2015) as our baseline to implement E-CRF. The detailed information follows standard settings in (Chen et al., 2014; 2018a) and we add it into our *Appendix A*. Specially, we employ SLIC (Achanta et al., 2012), a common superpixel segmentation algorithm, to divide each image of ADE20K, Cityscapes and Pascal Context into 200, 600, and 200 blocks respectively. Note that the superpixel is generated offline. To verify the effectiveness of our approach for semantic segmentation, we adopt two common metrics in our experiments, *i.e.*, class-wise mIoU to measure the overall segmentation performance and 1-pixel boundary F-score (Takikawa et al., 2019; Tan et al., 2023) to measure the boundary segmentation performance.

### 4.2 ABLATION STUDY

#### 4.2.1 COMPARISONS WITH RELATED METHODS

As shown in Table 1, we compare our proposed E-CRF with other traditional CRF-based methods, *i.e.*, Vanilla-CRF and Joint-CRF. First of all, it is clear that all the CRF-based methods outperform the baseline model by a large margin, which well verifies the main claim in (Chen et al., 2014; 2017a; Lin et al., 2015; Liu et al., 2015; Lin et al., 2016) that CRF is beneficial to boundary segmentation (F-score). What's more, E-CRF achieves the best result among all those methods, which surpasses the baseline model with up to **1.48%** mIoU and **2.20%** F-score improvements. E-CRF fuses the CRF mechanism into the CNN network as an organic whole. It relieves the disturbing gradients caused

Table 1: Comparisons with baseline, Vanilla-CRF and Joint-CRF on ADE20K *val* dataset. [3]It stands for DeeplabV3+ followed by DenseCRF. [4]An end-to-end manner of Vanilla-CRF, similar to (Zheng et al., 2015).

| Method | ResNet-50 | | ResNet-101 | |
|---|---|---|---|---|
| | F-score (%) | mIoU (%) | F-score (%) | mIoU (%) |
| DeeplabV3+ | 14.25 | 42.72 | 16.15 | 44.60 |
| Vanilla-CRF [3] | 16.26 | 43.18 (+0.46) | 17.89 | 45.14 (+0.54) |
| Joint-CRF [4] | 16.32 | 43.69 (+0.96) | 18.03 | 45.61 (+1.01) |
| **E-CRF (Ours)** | **16.45** | **44.20 (+1.48)** | **18.32** | **46.02 (+1.42)** |

by the BCWC problem and adjusts the feature representations to boost the overall segmentation performance and the boundary segmentation. Fig 1(a) also proves that E-CRF can decrease the inter-class similarity consistently which results in more discriminative feature representations. Experiments on Cityscapes dataset can be found in our *Appendix A*.

### 4.2.2 Ablation on Message Passing Strategies

As we have discussed in Sec. 3.2, two message passing components, *i.e.*, pairwise module $\psi_p^f$ and superpixel-based module $\psi_s^f$, play vital roles in our proposed E-CRF. Table 2 shows that $\psi_p^f$ and $\psi_s^f$ can boost the overall segmentation performance on ADE20K *val* dataset with up to **1.19%** mIoU and **1.25%** mIoU gains when integrated into the baseline model respectively. Moreover, if we fuse them as a whole into E-CRF, they can further promote the segmentation performance by up to **1.48%** mIoU improvements. We also compare with Non-local (Wang et al., 2018), another famous attention-based message passing method, into our experiments for comprehensive comparisons even though it actually has different design concepts from ours. Unfortunately, we find that although Non-local achieves improvements over the baseline, it is still inferior to our E-CRF.

Table 2: Comparisons between message passing strategies, and ablation studies for different message passing components in E-CRF, pairwise $\psi_p^f$ and auxiliary superpixel-based $\psi_s^f$.

| Method | $\psi_p^f$ | $\psi_s^f$ | mIoU(%) | |
|---|---|---|---|---|
| | | | ResNet-50 | ResNet-101 |
| DeeplabV3+ | | | 42.72 | 44.60 |
| + Non-local | | | 43.52 (↑ 0.80) | 45.34 (↑ 0.74) |
| **E-CRF** | ✓ | | 43.91 (↑ 1.19) | 45.47 (↑ 0.87) |
| | | ✓ | 43.83 (↑ 1.11) | 45.85 (↑ 1.25) |
| | ✓ | ✓ | **44.20 (↑ 1.48)** | **46.02 (↑ 1.42)** |

### 4.2.3 Ablation of Superpixel Numbers

We follow standard settings in our paper and take DeeplabV3+ based on ResNet-50 as the baseline model to present the performance of E-CRF under different superpixel numbers. Detailed comparisons on ADE20K dataset are reported in Table 3 and **SP** denotes **S**uper**P**ixel. As shown in Table 3, different numbers of superpixels indeed affect the performance of E-CRF. Intuitively, when the number of superpixels is 200, E-CRF acquires the best performance as it achieves a better trade-off between the superpixel purity and the long-range dependency. Moreover, it is worth noting that when the pairwise message passing strategy (*i.e.*, $\psi_p^f$) is also adopted in E-CRF, it becomes more robust to the different numbers of superpixels that may introduce noise, as our adaptive message passing mechanism (including $\psi_p^f$ and $\psi_s^f$) can be compatible with the variance.

Table 3: Comparisons with different superpixel numbers on ADE20K *val* dataset.

| SP num | mIoU w\ $\psi_p^f$ (%) | mIoU w\o $\psi_p^f$ (%) |
|---|---|---|
| No | 43.91 | 42.72 |
| 100 | 44.02 | 43.43 |
| **200** | **44.20** | **43.83** |
| 300 | 44.13 | 43.56 |
| 400 | 43.96 | 43.22 |

More ablation studies including comparison of different boundary refinement and computational cost are presented in Appendix A.4.

Table 4: Comparisons with other state-of-the-art methods on ADE20K *val* dataset, Cityscapes *val* and *test*, and Pascal Context *val* dataset.

| Method | backbone | mIoU(%) | | | |
|---|---|---|---|---|---|
| | | ADE-*val* | City-*val* | City-*test* | Pas-Con |
| CCNet (Huang et al., 2019) | ResNet101 | 45.22 | 81.3 | 81.9 | - |
| ANL (Zhu et al., 2019) | ResNet101 | 45.24 | - | - | 52.8 |
| GFFNet (Li et al., 2020c) | ResNet101 | 45.33 | 81.8 | 82.3 | 54.2 |
| APCNet (He et al., 2019b) | ResNet101 | 45.38 | - | - | 54.7 |
| DMNet (He et al., 2019a) | ResNet101 | 45.50 | - | - | 54.4 |
| SpyGR (Li et al., 2020a) | ResNet101 | - | 80.5 | 81.6 | 52.8 |
| RecoNet (Chen et al., 2020) | ResNet101 | 45.54 | 81.6 | 82.3 | 54.8 |
| SPNet (Hou et al., 2020) | ResNet101 | 45.60 | - | - | 54.5 |
| DNL (Yin et al., 2020) | ResNet101 | 45.82 | - | - | 55.3 |
| RANet (Shen et al., 2020) | ResNet101 | - | 81.9 | 82.4 | 54.9 |
| ACNet (Fu et al., 2019) | ResNet101 | 45.90 | 82.0 | 82.3 | 54.1 |
| HANet (Choi et al., 2020) | ResNet101 | - | 82.05 | 82.1 | - |
| RPCNet (Zhen et al., 2020b) | ResNet101 | - | 82.1 | 81.8 | - |
| CaCNet (Liu et al., 2020) | ResNet101 | 46.12 | - | - | 55.4 |
| CPNet (Yu et al., 2020a) | ResNet101 | 46.27 | - | - | 53.9 |
| STLNet (Zhu et al., 2021) | ResNet101 | 46.48 | 82.3 | 82.3 | 55.6 |
| **E-CRF (Ours)** | ResNet101 | **46.83** | **82.74** | **82.5** | **56.1** |

## 4.3 COMPARISONS WITH SOTA METHODS

*In this research, we mainly focus on the Boundary-caused Class Weight Confusion (BCWC) in CNN models. Hence, in this section, we choose CNN-based methods for fair comparisons.*[5]

**ADE20K.** We first compare our E-CRF (ResNet101 as backbone) with existing methods on the ADE20K *val* set. We follow standard settings in (Huang et al., 2019; Yuan et al., 2020a; Zhu et al., 2021) to adopt multi-scale testing and left-right flipping strategies. Results are presented in Table 4. It is shown that E-CRF outperforms existing approaches. Segmentation visualization is presented in our *Appendix*

**Cityscapes.** To verify the generalization of our method, we perform detailed comparisons with other SOTA methods on Cityscapes *val* and *test* set. Multi-scale testing and left-right flipping strategies are also adopted. The results with ResNet101 as backbone are reported in Table 4. Remarkably, our algorithm achieves **82.74%** mIoU in val set and outperforms previous methods by a large margin.

**Pascal Context.** To further verify the generalization of E-CRF (ResNet101 as backbone), we compare our method with other SOTA method on Pascal Context dataset as shown in Table 4. We adopt multi-scale testing and left-right flipping strategies as well. The result suggests the superiority of our method.

## 5 CONCLUSION AND FUTURE WORKS

In this paper, we focus on the particularity of class weights in semantic segmentation and explicitly consider an important issue , named as *Boundary-caused Class Weights Confusion* (BCWC). We dive deep into it and propose a novel method, *E-CRF*, via combining CNN network with CRF as an organic whole to alleviate BCWC from two aspects (*i.e.*, scale and direction). In addition, we make an exhaustive theoretical analysis to prove the effectiveness of E-CRF. Eventually, our proposed method achieves new results on ADE20K, Cityscapes, and Pascal Context datasets.

There are two important directions for future research. In this work, we use SLIC, a common cluster-based algorithm for fast implementation. There exist many other superpixel algorithms such as graphical-based (Felzenszwalb & Huttenlocher, 2004) and CNN-based (Jampani et al., 2018) that may give better boundary results for objects. Therefore how these different methods influence the performance in our framework is interesting. Besides, We find that transformer-based networks suffer from BCWC issue as well and make a preliminary exploration. More works are expected to focus on this issue.

---

[5]We also conduct experiments based on SegFormer (Xie et al., 2021) to make a preliminary exploration in our Appendix B as we found that BCWC issue also exists in transformer-based models.

## ACKNOWLEDGMENTS

We thank the anonymous reviewers for their constructive comments. We also sincerely thank Tiancai Wang for useful discussion. This work is supported by the NSFC Grants no. 61972008.

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

# A APPENDIX

## A.1 FORMULA DERIVATION

Firstly, we give the gradient equation of $\nabla W_c$ mentioned in this paper:

$$\nabla W_c = \frac{\partial L_k}{\partial W_c} = \frac{\partial L_k}{\partial P_k^c} \cdot \frac{\partial P_k^c}{\partial Y_k^c} \cdot \frac{\partial Y_k^c}{\partial W_c}. \tag{16}$$

According to it, we present the derivative of each term respectively:

$$\frac{\partial L_k}{\partial P_k^c} = -\frac{1}{P_k^c}, \tag{17}$$

$$\frac{\partial P_k^c}{\partial Y_k^c} = \frac{((\sum_{m \neq c} e^{Y_k^m}) + e^{Y_k^c}) \cdot e^{Y_k^c} - (e^{Y_k^c})^2}{((\sum_{m \neq c} e^{Y_k^m}) + e^{Y_k^c})^2} = \frac{e^{Y_k^c}}{(\sum_{m \neq c} e^{Y_k^m}) + e^{Y_k^c}} - (\frac{e^{Y_k^c}}{(\sum_{m \neq c} e^{Y_k^m}) + e^{Y_k^c}})^2, \tag{18}$$

and

$$\frac{\partial Y_k^c}{\partial W_c} = F_k. \tag{19}$$

Further, it is worth noting that $P_k^c$ is given by:

$$P_k^c = \frac{e^{Y_k^c}}{(\sum_{m \neq c} e^{Y_k^m}) + e^{Y_k^c}}. \tag{20}$$

To simplify Eq (18), we take Eq (20) into account. Thus, Eq (18) is formulated as:

$$\frac{\partial P_k^c}{\partial Y_k^c} = P_k^c - P_k^c \cdot P_k^c = P_k^c \cdot (1 - P_k^c). \tag{21}$$

So after integrating Eq (17), Eq (21), and Eq (19), Eq (16) can be formulated as:

$$\nabla W_c = \frac{\partial L_k}{\partial W_c} = -\frac{1}{P_k^c} \cdot P_k^c \cdot (1 - P_k^c) \cdot F_k = -(1 - P_k^c) \cdot F_k. \tag{22}$$

Finally, the class weights $W_c^*$ in the next iteration will be updated:

$$W_c^* = W_c - \nabla W_c = W_c + (1 - P_k^c) \cdot F_k. \tag{23}$$

## A.2 EXPERIMENT SETUP

**ADE20K** ADE20K (Zhou et al., 2017) is one of the most challenging benchmarks, containing 150 fine-grained semantic concepts and a variety of scenes with 1,038 image-level labels. There are 20210 images in training set and 2000 images in validation set.

**Cityscapes** Cityscapes (Cordts et al., 2016) has 5,000 images captured from 50 different cities. Each image has $2048 \times 1024$ pixels, which have high quality pixel-level labels of 19 semantic classes. There are 2,975 images in training set, 500 images in validation set and 1,525 images in test set. We do not use coarse data in our experiments.

**Pascal Context** PASCAL Context (Mottaghi et al., 2014) is a challenging scene understanding dataset, which provides the semantic labels for the images. There are 4, 998 images for training and 5, 105 images for validation on PASCAL Context dataset. In our experiment, the 59 most frequent categories are used for training.

**Implementation Details.** The initial learning rate is set as 0.01 for both datasets. We employ a poly learning rate strategy where the initial learning rate is multiplied by $(1 - iter/totaliter)^{0.9}$ after each iteration. We set training time to 80000 iterations for ADE20K and Pascal Context, and 180 epochs for Cityscapes. Momentum and weight decay coefficients are set as 0.9 and 0.0005, respectively. For data augmentation, we apply the common scale (0.5 to 2.0), cropping and flipping of the image to augment the training data. Input size for ADE20K dataset is set to $512 \times 512$, and $480 \times 480$ is for Pascal Context while input size for Cityscapes dataset is set to $832 \times 832$. The syncBN (Peng et al., 2018) is adopted in all experiments, and batch size on ADE20K and Pascal Context is set to 16 and it is set to 8 for Cityscapes.

## A.3  Comparisons with Related Methods on Cityscapes

To further evaluate our proposed method, we thoroughly compare our approach with baseline and other traditional CRF-based methods, *i.e.*, Vanilla-CRF and Joint-CRF on Cityscapes dataset. As shown in Table 5, E-CRF achieves the best result among all those methods, which outperforms the baseline model by **0.92%** in mIoU and **3.81%** in F-score respectively. Obviously, our method is more effective than both Vanilla-CRF and Joint-CRF.

| Method | ResNet-50 | | ResNet-101 | |
|---|---|---|---|---|
| | F-score (%) | mIoU (%) | F-score (%) | mIoU (%) |
| DeeplabV3+ | 60.48 | 79.54 | 61.94 | 80.85 |
| Vanilla-CRF | 62.38 | 79.65 (+0.11) | 63.46 | 80.92 (+0.07) |
| Joint-CRF | 63.44 | 79.78 (+0.24) | 64.43 | 81.05 (+0.20) |
| **E-CRF (Ours)** | **64.29** | **80.35 (+0.81)** | **65.57** | **81.77 (+0.92)** |

Table 5: Comparisons with baseline, Valina-CRF and Joint-CRF on Cityscapes *val* dataset.

## A.4  More Ablation Studies

### A.4.1  Different Boundary Refinement

We consider three typical methods inlcuding Segfix Yuan et al. (2020b), DecoupleSegNet Li et al. (2020b), and ABL [6] (Wang et al., 2022). They refine boundary segmentation via post-processing, improving boundary representation, and adding boundary allignment loss respectively. But all of them ignore the existence of BCWC issue, which may restrict their capability. In Table 4, we use DeeplabV3+ with ResNet101 as our baseline method. For Segfix, we use the official code (It uses HRNet as backbone) to boost the performance of DeeplabV3+. For DecoupleSegNet which is constructed based on DeeplabV3+, we also use the official code (It uses ResNet101 as backbone). All the models are trained on ADE20K for 80K iterations with batch size set to 16. When testing, we adopt the single-scale testing strategy (i.e., raw image) because using a single scale (i.e., raw images) when comparing with baselines or performing ablation studies is a traditional default setting in the semantic segmentation field. The goal is to eliminate the effect of other elements (e.g., image augmentation).

It is observed that all the methods enhance the performance and E-CRF achieves the highest mIoU and F-score. We speculate that this is because E-CRF has explicitly considered the BCWC problem and optimizes the class weights from both scale and direction aspects while refining boundary representation. It also indicates the importance of obtaining distinguishable class weights in semantic segmentation.

Table 6: Comparisons with other boundary refining methods on ADE20K *val* dataset.

| Method | mIoU (%) | F-score (%) |
|---|---|---|
| DeeplabV3+ (Chen et al., 2018a) | 44.60 | 16.15 |
| SegFix (Yuan et al., 2020b) | 45.62 | 18.14 |
| DecoupleSegNet (Li et al., 2020b) | 45.73 | 18.02 |
| ABL (Wang et al., 2022) | 45.38 | - |
| **E-CRF** | **46.02** | **18.32** |

### A.4.2  Comparisons on Computational Costs

We take DeeplabV3+ based on ResNet101 as the baseline model to perform the training time comparisons. Image size is set to $512 \times 512$ and all the experiments are conducted on 8 GeForce RTX 2080Ti GPUs with two images per GPU. The FLOPs, parameter size, and inference FPS are also reported in Table 7. We can find that our proposed E-CRF brings negligible extra costs over the baseline model. The cost difference between E-CRF and Joint-CRF is marginal. We also measure the time consuming of superpixel method (**5ms**), which is much smaller than that of inference (**55ms**).

---

[6]The authors do not provide the code in their paper. Hence, we just report the result based on OCRNet (Yuan et al., 2020a) in their paper.

Table 7: Comparisons on Computational costs on ADE20K dataset.

| Method | Backbone | Training Time(s) | FLOPs(G) | Parameters(M) | FPS |
|---|---|---|---|---|---|
| DeeplabV3+ | ResNet101 | 0.71 | 254.8 | 60.1 | 19.75 |
| Vanilla-CRF | ResNet101 | 0.71 | 254.8 | 60.1 | 1.86 |
| Joint-CRF | ResNet101 | 0.73 | 254.9 | 60.2 | 19.04 |
| E-CRF | ResNet101 | 0.74 | 255.0 | 60.2 | 18.32 |

## A.5 DISCUSSION

In this section, we discuss the difference between E-CRF and three related works including PC-Grad (Yu et al., 2020b), OCNet (Yuan & Wang, 2018), and SegFix (Yuan et al., 2020b).

**Difference between Projecting Conflicting Gradients (PCGrad) and E-CRF:** PCGrad and E-CRF are both gradient-based methods that focus on adjusting the gradient properly to optimize the learning process more effectively and efficiently. PCGrad is designed to mitigate a key optimization issue in multi-task learning caused by conflicting gradients, where gradients for different tasks point away from one another as measured by a negative inner product. If two gradients are conflicting, PCGrad alters the gradients by projecting each onto the normal plane of the other, preventing the interfering components of the gradient from being applied to the network. The idea behind PCGrad is simple and the method is effective. PCGrad is a task-level gradient optimization method, mainly focusing on conflicting gradients caused by multiple tasks during training (*e.g.*, in semantic segmentation and depth estimation). E-CRF is a finer-grained pixel-level gradient optimization method. E-CRF mainly aims at mitigating the boundary-caused class weights confusion in semantic segmentation via adjusting class weights from both scale and direction.

**Difference between OCNet and E-CRF:** OCNet uses self-attention to implement the object context pooling module. The object context pooling estimates the context representation of each pixel $i$ by aggregating the representations of the selected subset of pixels based on the estimated dense relation matrix. Further, OCNet combines context pooling module with the conventional multi-scale context schemes including PPM and ASPP. In E-CRF, we follow the idea behind Conditional Random Filed and embed it from logit space to deep-feature space. We instance the unary function and reformulate the pairwise function with a convolutional-based kernel. The kernel takes raw image RGB value and relative position embeddings as inputs (See Eq (6)), which is different from self-attention that takes extracted deep features as inputs. We also maintain one special term in CRF called label compatibility and transfer it to feature compatibility (See Eq (7)). Such is missing in self-attention. Besides, we do not measure the similarity between superpixel center and boundary pixels. We simply leverage the local prior in superpixel and use it to guide deep feature averaging. The motivation is to suppress noise information. Finally, the motivation between OCNet and E-CRF is different. OCNet mainly focuses on integrating as much object context as possible while E-CRF explicitly targets on the BCWC problem and optimizes class weights from both scale and direction.

**Difference between SegFix and E-CRF:** SegFix first encodes input image and predicts a boundary map and a direction map. Then SegFix uses the predicted boundary map and offset map derived from the direction map to correct the wrongly classified boundary pixels via internal points with high confidence. SegFix is beneficial for refining boundary segmentation and is served as a post-processing method, thus lacking the capability to alleviate BCWC problem. Our method is derived from traditional CRF (a post-processing method) and can be regarded as a plug-and-play module. It can be easily integrated with other methods. Besides, our method extends the optimization flexibility for BCWC problem. Empirically, we have compared Segfix and E-CRF based on DeeplabV3+ in Table 6 (Please see A.4.1). E-CRF produces 46.02% for mIoU on ADE20K, outperforming SegFix (45.62%) by 0.4%. This indicates the importance of alleviating BCWC problem.

## A.6 PAIRWISE MESSAGE PASSING VISUALIZATION

E-CRF takes an equivalent transformation to replace hand-designed Gaussian kernels in Vanilla-CRF with simple convolution operators for more flexible end-to-end optimization. The convolution operation involves two aspects. One is the appearance similarity and the other one is the relative position between pixels. As depicted in Fig 4(a), we take a pixel $k$ in the stool for an example and

show its relationship with other pixels in the image. Fig 4(b) and Fig 4(c) show its appearance similarity and *cosine* position embedding with other pixels respectively. It is clear that pixels share similar colors or close to the target pixel $k$ tend to be highlighted. Subsequently, in Fig 4(d), we directly visualize the results of our pairwise message passing module $\psi_p^f$ defined in Eq.(5). We can find that $\psi_p^f$ becomes concentrated on the most relevant pixels compared with the pixel $k$, which verifies the reliability of our pairwise message passing design.

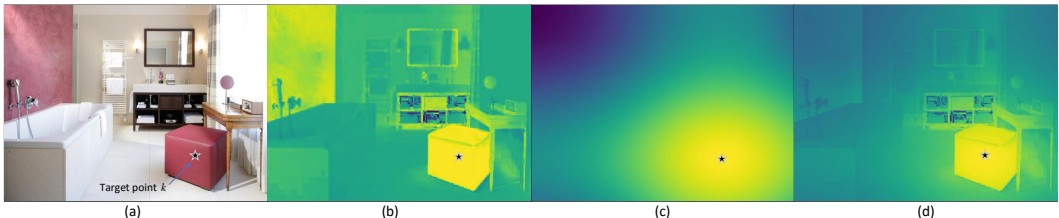

Figure 4: Visualization of pairwise message passing module $\psi_p^f$ in E-CRF. **(a)** A target pixel $k$ of the stool in the image. **(b)** The appearance similarity between other pixels and $k$. Pixels share the similar colors with $k$ tend to be highlighted. **(c)** The visualization of the relative position between $k$ and other pixels. Pixels close to $k$ achieve higher values. **(d)** The visualization of $\psi_p^f$ in E-CRF. $\psi_p^f$ focuses more on most relevant pixels compared to $k$.

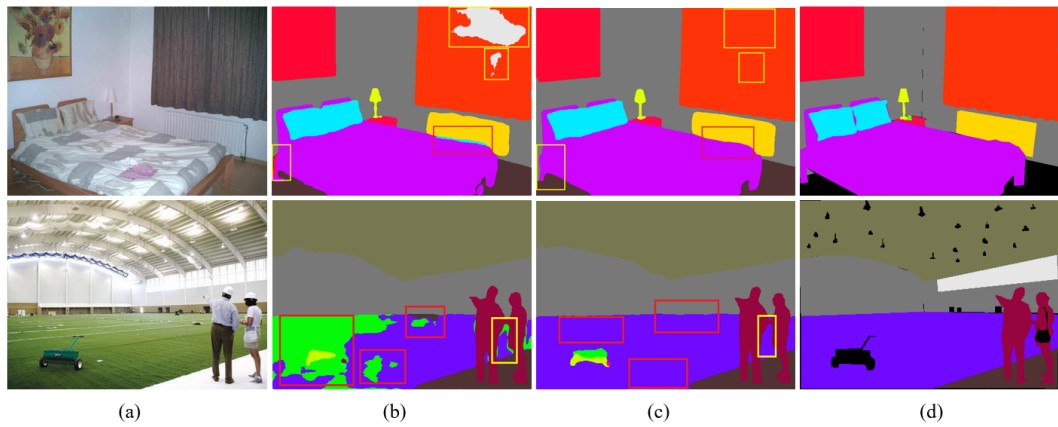

Figure 5: Visualization comparisons between our method and baseline on ADE20K validation set. **(a)** Images from ADE20K dataset. **(b)** Segmentation output from DeeplabV3+. **(c)** Segmentation output from our method. Obviously, compared with baseline, the results are segmented well by E-CRF. **(d)** Image labels.

## B    BCWC IN TRANSFORMER

Are transformer-based models also suffering from Boundary-caused Class Weight Confusion? Is our method effective to transformer-based models? To answer these questions, we make a preliminary exploration in this section.

### B.1    OBSERVATIONS ON ADE20K

Following the same idea in Fig.1(a) of this paper, we take Segformer (Xie et al., 2021) (a transformer-based segmentation model) as an example to train on ADE20K (Zhou et al., 2017) dataset. We count the number of adjacent pixels for each class pair and find a corresponding category that has the most adjacent pixels for each class. Then, we calculate the similarity of their class weights and depict it in Fig 6. X-axis stands for the number of adjacent pixels for each class pair in descending order, and Y-axis represents the similarity of their class weights. Blue line denotes Segformer while orange line denotes E-CRF based on Segformer. As shown in Fig 6, two categories that share more adjacent pixels are inclined to have more similar class weights, while E-CRF effectively decreases the similarity between adjacent categories and makes their class weights more discriminative. These observations on transformer-based model are quite similar to previous results in CNN-based models. Apparently, transformer-based models are also suffering from Boundary-caused Class Weight Confusion.

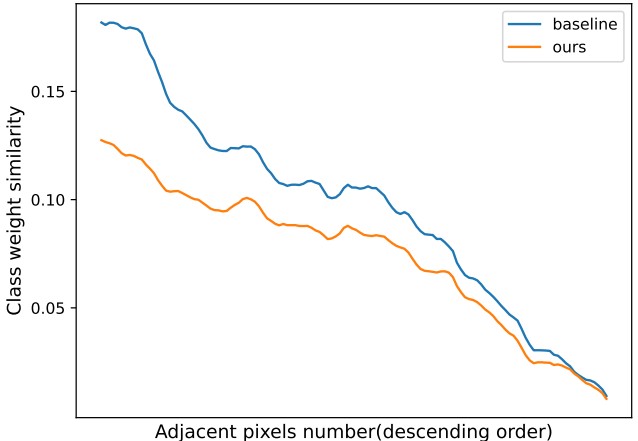

Figure 6: Class weight similarity on transformer-based model

### B.2    EFFECTIVENESS ON TRANSFORMER

To evaluate the effectiveness of our method, we take Segformer (Xie et al., 2021) (based on MiT-B5) as our transformer baseline and incoporate E-CRF into it. Experiments are conducted on ADE20K and Cityscapes datasets. Similarly, we also compare our method with other traditial CRF-based methods, *i.e.*, Vanilla-CRF and Joint-CRF [7]. As shown in Table 8, E-CRF achieves the best result among all those methods, which surpasses the baseline model with up to **1.01%** mIoU and **3.81%** F-score improvements. By E-CRF relieves the disturbing gradients caused by the BCWC problem boost the overall segmentation performance and the boundary segmentation. Fig 6 also proves that E-CRF can decrease the inter-class similarity consistently which results in more discriminative feature representations.

| Method | ADE20K | | Cityscapes | |
|---|---|---|---|---|
| | F-score (%) | mIoU (%) | F-score (%) | mIoU (%) |
| Segformer | 18.53 | 49.13 | 62.42 | 82.25 |
| Vanilla-CRF | 21.72 | 49.36 (+0.23) | 64.06 | 82.31 (+0.06) |
| Joint-CRF | 21.91 | 49.55 (+0.42) | 64.93 | 82.41 (+0.16) |
| **E-CRF (Ours)** | **22.34** | **50.14 (+1.01)** | **66.05** | **83.07 (+0.82)** |

Table 8: Comparisons with baseline, Valina-CRF, and Joint-CRF on ADE20K and Cityscapes *val* datasets.

---

[7]Note that they are also based on Segformer

## B.3 COMPARISONS WITH SOTA METHODS

To further verify the effectiveness, we compare our methods with other transformer-based SOTA methods with similar number of parameters (except SETR) for fair comparisons in both ADE20K and Cityscapes datasets. Multi-scale testing and left-right flipping strategies are adopted. As shown in Table 9, our method achieves the best results among all the SOTA methods in both ADE20K and Cityscapes datasets. Besides, our method also has the smallest number of parameters.

Table 9: Comparisons with other transformer-based SOTA methods on ADE20K *val* dataset and Cityscapes *val* and *test* dataset.

| Method | Backbone | mIoU(%) | | | Params (M) |
|---|---|---|---|---|---|
| | | ADE-*val* | City-*val* | City-*test* | |
| SETR (Zheng et al., 2021) | ViT-L (307M) | 50.20 | 82.15 | 82.2 | 310M |
| UperNet | Swin-B (Liu et al., 2021) (88M) | 49.65 | - | - | 121M |
| UperNet | Twins-L (Chu et al., 2021) (99M) | 50.20 | - | - | 133M |
| SegFormer (Xie et al., 2021) | MiT-B5 (81M) | 50.22 | 83.48 | 82.2 | 85M |
| UperNet | XCiT-M24 (Chu et al., 2021) (84M) | 48.40 | - | - | 109M |
| DPT (Ranftl et al., 2021) | ViT-B (86M) | 48.34 | - | - | 112M |
| Segmentor (Strudel et al., 2021) | DeiT-B (86M) | 50.08 | 80.60 | - | 86M |
| **E-CRF (Ours)** | MiT-B5 (81M) | **51.28** | **83.7** | **82.5** | 85M |

