# OpenReview forum: "E-CRF: Embedded Conditional Random Field for Boundary-caused Class Weights Confusion in Semantic Segmentation"
_ICLR.cc/2023/Conference — ICLR 2023 poster_

### Official Review · Reviewer_VZu5 · 2022-10-23

**Confidence:** 1
**Clarity, Quality, Novelty And Reproducibility:** yes.
**Correctness:** 4
**Technical Novelty And Significance:** 2
**Empirical Novelty And Significance:** 2
**Recommendation:** 5

**Strength And Weaknesses:**

Strength:
1. The paper is well-written and easy to understand.
2. The idea Embedded CRF (E-CRF) is used to address the BCWC problem is good.
3. The experiments shows the proposed methods are effective and verified.


Weakness
1. The proposed E-CRF is very similar to Joint-CRF. Though the authors claims that joint-crf  lacks the ability to optimize class representations effectively due to the defective design and E-CRF can solve it well, the experimental results can not show E-CRF improve a lot.  In table 1, E-crf has only no more than 1% improvement than joint-crf. This improvement is not so obvious.
2. E-CRF adopts superpixel  as an auxiliary to further strengthen the reliability of the message passing to the boundary pixels. Compared with joint-crf, it need more computation for superpixel generation. That is to say, E-CRF use more computation resourses.
3. There is no model size comparison and time cost comparison.


**Summary Of The Paper:**

This paper focus on the particularity of class weights in semantic segmentation and explicitly consider an important issue , named as Boundary-caused Class Weights Confusion (BCWC).  The authors propose a novel method, E-CRF, via combining CNN network with CRF as an organic whole to alleviate BCWC from two aspects (i.e., scale and direction). In addition, the authors make an exhaustive theoretical analysis to prove the effectiveness of E-CRF. Eventually, the proposed method achieves new results on ADE20K, Cityscapes, and Pascal Context datasets.

**Summary Of The Review:**

Though this paper idea is good, but we cannot see the experiment results are convincing enough. As i put in the *Strength And Weaknesses, my main concern is that the experimental results between E-CRF and joint-crf.

---

> ### Author Response · Authors · 2022-11-13
> **Thanks and response to Reviewer VZu5**
>
> **Q1:** Though the authors claims that joint-crf lacks the ability to optimize class representations effectively due to the defective design and E-CRF can solve it well, the experimental results can not show E-CRF improve a lot. In table 1, E-crf has only no more than 1% improvement than joint-crf. This improvement is not so obvious.
>
> **A1:** **Though this improvement is not so obvious, it is non-trivial. Two reasons are listed below.**
>
> Firstly, the ADE20K dataset contains a total of 150 categories and is the most difficult dataset in the semantic segmentation field due to complicated scenes and diverse categories.
>
> Secondly, considering other methods published at various top conferences (e.g., CVPR, NeurIPS, ICCV, and ECCV) in Table 6, 0.5% mIoU gain is non-trivial. For example, on ADE20K, STLNet (CVPR2021) is 0.21% higher in mIoU than CPNet (CVPR2020). RecoNet (ECCV2020) is 0.16% higher in mIoU than APCNet (ICCV2019).
> Please see Sec 4.4.
>
> **Q2:** E-CRF adopts superpixel as an auxiliary to further strengthen the reliability of the message passing to the boundary pixels. Compared with joint-crf, it need more computation for superpixel generation. That is to say, E-CRF use more computation resourses.
>
> **A2:** For training, the superpixel is prepared offline. Please see 4.1. So the only cost is to load the superpixel during training. Such cost is negligible. For testing, we measure the time consumption of superpixel method (about 5ms). It is one-eleventh of inference time consumption (about 55ms). Please see Sec 4.3.
>
> **Q3:** There is no model size comparison and time cost comparison.
>
> **A3:** We have compared the model size and time cost comparison in Table 5. Please see Table 5. We also provide the comparison among Vanilla-CRF, Joint-CRF, and E-CRF below on ADE20K. The cost difference between E-CRF and Joint-CRF is marginal.
>
> | Method | Backbone | Training Time(s) | FLOPs(G) | Parameters(M) | FPS |
> | --- | --- | --- | --- | --- | --- |
> | DeeplabV3+ | ResNet101 | 0.71 | 254.8 | 60.1 | 19.75 |
> | Vallina-CRF | ResNet101 | 0.71 | 254.8 | 60.1 | 1.86 |
> | Joint-CRF | ResNet101 | 0.73 | 254.9 | 60.2 | 19.04 |
> | E-CRF | ResNet101 | 0.74 | 255.0 | 60.2 | 18.32 |

---

> ### Author Response · Authors · 2022-12-03
> **Looking forward to more discussions !**
>
> Dear Reviewer VZu5:
>
> We sincerely appreciate your time and efforts in reviewing our paper, which would help us improve our final paper!
>
> As the deadline for discussion is approaching, please don’t hesitate to let us know if there are any additional clarifications we can offer.
>
> Look forward to your post-rebuttal rating!

---

### Official Review · Reviewer_aDCi · 2022-10-24

**Confidence:** 5
**Correctness:** 4
**Technical Novelty And Significance:** 2
**Empirical Novelty And Significance:** 2
**Recommendation:** 6

**Clarity, Quality, Novelty And Reproducibility:**

Clarity and Quality:
This paper is well written with clear definition and explanation.

Novelty:
The proposed solution of dealing with BCWC is not novel, but the theoretical analysis is insightful.

Reproducibility:
It is clear to reproduce the method.


**Details Of Ethics Concerns:**

No ethics concerns.

**Strength And Weaknesses:**

Strength:
+ This paper is well-written and provides insights, e.g., the illustration and interpretation of optimization of different CRF schemes is insightful.
+ The proposed method extends the gradient descent from scale aspect to both scale and direction aspects, making the optimization more effective.


Weakness

- The discussed solution on CRF is a little narrow and out-of-date. Since class-weight problem is general for all semantic segmentation models, it could be better to extend the method to other models.  I also look forward to a discussion part for how to extend this method to Transformer-based segmentation models.
- Although the discussion of BCWC is meaningful, the proposed solution is not novel. The method of discriminating boundary pixels from local representative pixels (e.g., super-pixel center) by measuring their similarities (attention) is commonly used. Similar ideas are in the below papers
  - OCNet: Object Context for Semantic Segmentation,
  - SegFix: Model-Agnostic Boundary Refinement for Segmentation
- The improvement from Joint-CRF is marginal, about 0.5% mIoU.
- The design of using super-pixel to extract pixel representation is not elegant. It takes additional cost, and the segmentation performance highly depends on the quality of super-pixel. Is there any way to embed this process into the model, to make the partitioned representation learnable?



**Summary Of The Paper:**

This paper addresses the Boundary-caused Weights Confusion problem in semantic segmentation. The authors propose an E-CRF model by integrating CRF into CNN and pursuing gradient descent not only from scale but direction aspects. The method improves baseline for about 1% mIoU.

**Summary Of The Review:**

This paper focuses on addressing BCWC problem in CRF, which could have been more generic in class weighting problem of semantic segmentation. The analysis is insightful and interesting, but the proposed solution is not novel and elegant, commonly used and taking additional cost. I consider this paper could be better if it extends more, but currently it lacks novelty in solution.

---

> ### Author Response · Authors · 2022-11-13
> **Thanks and response to Reviewer  aDCi (2)**
>
> **Q4:** The improvement from Joint-CRF is marginal, about 0.5% mIoU.
>
> **A4:** **We need to explain that the improvement is non-trivial. There are two reasons.**
>
> Firstly, the mIoU gain is obtained over 150 categories in ADE20K. In the semantic segmentation field, the ADE20K dataset is the most difficult dataset due to its complicated scenes and diverse categories.
>
> Secondly, considering other methods published at various top conferences (e.g., CVPR, NeurIPS, ICCV, and ECCV) in Table 6, 0.5% mIoU gain is non-trivial. For example, STLNet (CVPR2021) is 0.21% higher in mIoU than CPNet (CVPR2020). RecoNet (ECCV2020) is 0.16% higher in mIoU than APCNet (ICCV2019).
> Please see Sec 4.4.
>
> **Q5:** The design of using super-pixel to extract pixel representation is not elegant. It takes additional cost, and the segmentation performance highly depends on the quality of super-pixel. Is there any way to embed this process into the model, to make the partitioned representation learnable?
>
> **A5:** Thanks for your interest in our method. Is there any way to embed this process into the model, to make the partitioned representation learnable? Yes is our answer. In this work, we just choose SLIC (a cluster-based superpixel algorithm) for fast implementation and mainly aim to verify the feasibility and effectiveness. To the best of our knowledge, there exist CNN-based learnable superpixel generation networks such as Superpixel Sampling Networks (SSN) (ECCV2020). And combining SSN or similar networks with our method is feasible theoretically and is also likely to create a more general framework. However, some problems may be raised in that way, e.g., how to combine the backbone network of SSN with our network properly (SSN does not use ResNet) and how to balance the loss between SSN loss and segmentation loss. Besides, it will also introduce more parameters and increase model size. We have considered it as future work in Sec 5. Please see Sec 6.

---

> ### Author Response · Authors · 2022-11-13
> **Thanks and response to Reviewer  aDCi (1)**
>
> **Q1:** Since class-weight problem is general for all semantic segmentation models, it could be better to extend the method to other models.
>
> **A1:** Thanks for your kind advice. We take CCNet (43.97% mIoU on ADE20K;TPAMI 2020) and OCNet (44.28% mIoU on ADE20K;IJCV2021) with backbone ResNet101 as our baselines in single-scale testing. When equipped with our method following the same setting in our paper, we produce 45.18% and 45.59% for mIoU, outperforming CCNet and OCNet by 1.21% and 1.31% respectively.
>
> **Q2:** A discussion part for how to extend this method to Transformer-based segmentation models.
>
> **A2:** It is easy to extend our method to Transformer-based segmentation models **because our method is simply embedded before the final classification layer (1X1 Conv) and can be regarded as a plug-and-play module.** Concretely, we do not need to change the whole architecture of transformer-based segmentation models. The only thing we need to do is just to add our module right before the final classification layer (1X1 Conv) of transformer-based segmentation models. In fact, we indeed conduct related experiments on SegFormer (NeurIPS2021 a transformer-based segmentation model) and obtain a 1.01% improvement on ADE20K. Please see Appendix B.
>
> **Q3:** Although the discussion of BCWC is meaningful, the proposed solution is not novel. The method of discriminating boundary pixels from local representative pixels (e.g., super-pixel center) by measuring their similarities (attention) is commonly used. Similar ideas are in OCNet and SegFix.
>
> **A3:**  We thank the reviewer for admitting the meaningfulness of our work. We want to point out the differences between our E-CRF and the two methods, i.e., OCNet and SegFix.
>
> **Difference between OCNet and E-CRF:** OCNet uses self-attention to implement the object context pooling module. The object context pooling estimates the context representation of each pixel$i$by aggregating the representations of the selected subset of pixels based on the estimated dense relation matrix. Further, OCNet combines context pooling module with the conventional multi-scale context schemes including PPM and ASPP. In E-CRF, we follow the idea behind Conditional Random Filed and embed it from logit space to deep-feature space. We instance the unary function and reformulate the pairwise function with a convolutional-based kernel. The kernel takes raw image RGB value and relative position embeddings as inputs (See equation (6)), which is different from self-attention that takes extracted deep features as inputs. We also maintain one special term in CRF called label compatibility and transfer it to feature compatibility (See equation (7)). Such is missing in self-attention. Besides, we do not measure the similarity between superpixel center and boundary pixels. We simply leverage the local prior in superpixel and use it to guide deep feature averaging. The motivation is to suppress noise information. Finally, the motivation between OCNet and E-CRF is different. OCNet mainly focuses on integrating as much object context as possible while E-CRF explicitly targets on the BCWC problem and optimizes class weights from both scale and direction. Empirically, we combine our method with OCNet and produce 45.59% for mIoU on ADE20K, outperforming OCNet(44.28%) by 1.31%. This indicates that our method is also complementary with OCNet. Please see answer A1.
>
> **Difference between SegFix and E-CRF:** SegFix first encodes input image and predicts a boundary map and a direction map. Then SegFix uses the predicted boundary map and offset map derived from the direction map to correct the wrongly classified boundary pixels via internal points with high confidence. SegFix is beneficial for refining boundary segmentation and is served as a post-processing method, thus lacking the capability to alleviate BCWC problem. Our method is derived from traditional CRF (a post-processing method) and can be regarded as a plug-and-play module. It can be easily integrated with other methods. Our experiments on CNN-based methods e.g., CCNet and OCNet (Please see answer A1) and transformer-based methods e.g., Segformer (Please see Appendix B and answer A2) verify this. Besides, our method extends the optimization flexibility for BCWC problem. Empirically, we have compared Segfix and E-CRF based on DeeplabV3+ in Table 4 (Please see Sec 4.2.4). E-CRF produces 46.02% for mIoU on ADE20K, outperforming SegFix (45.62%) by 0.4%. This indicates the importance of alleviating BCWC problem.
>
> We are willing to cite the two works i.e., OCNet and SegFix, and add this discussion in our final paper.

---

### Official Review · Reviewer_qXNf · 2022-10-25

**Confidence:** 5
**Correctness:** 3
**Technical Novelty And Significance:** 3
**Empirical Novelty And Significance:** 2
**Recommendation:** 6

**Clarity, Quality, Novelty And Reproducibility:**


Clarity
I think the paper has not fully made itself clear about how to generalize the proposed approach for addressing the multi-class classification case. The current formulation only touches the binary case. This needs to be clarified.

Quality
The paper is presented in a good quality. However, I think a more slice and dice results need to be presented to back up the claims in the contribution of the paper.


Novelty
I think the proposed approach is novel, though it needs to ensure that the previous literatures are properly referred and discussed.


Reproducibility
The proposed approach seems to be easy to understand and could be reproduced with the third parties.



**Strength And Weaknesses:**

Pros
The main contribution of the proposed E-CRF is to introducing the perspective of gradient descent and optimizes the class weights using both scale and direction using back propagation. This seems to be novel to me as I did not find any other similar semantic segmentation works touch this perspective. Recent works either focus on updating the backbone architectures, or on how to do better than the CRF message passing. This paper brings the new perspective of different gradient-based method.

The proposed approach has backed the claims with impressive results on various challenging semantic segmentation datasets including ADE20K, Pascal Context, and Cityscapes. The improvements are relative small, but the gain is built on top of the previous state-of-the-art methods.

Cons
I think the main weakness is that the paper does not really refer the good literature, making it appears that the gradient-based method is introduced the first time and the authors apply that to the semantic segmentation. In particularly, I would consider the proposed approach is very similar to the other works such as PCGrad (Gradient Surgery for Multi-Task Learning, https://arxiv.org/abs/2001.06782, NeurIPS 2020), while the paper does not cite that one.

Another issue is that the proposed approach does not provide sufficient experiments to support the claim. I would like to see the fine-grained experiments to show that the proposed approach get real improvements on the classes that are sharing the most adjacent pixels. This will be like a slice-and-dice results from the aggregated results reported by the current experiments.

Third, I am not fully understand how the proposed approach generalize the formulation to the multi-class classification cases. The current formulation presented in the paper only covers the binary case.

**Summary Of The Paper:**

The goal of this paper is to improve the semantic segmentation performance through addressing the boundary-caused class weights confusion issue. The two semantic classes that share more adjacent pixels tend to have more similar class weights in the state-of-the-art semantic segmentation solution such as DeepLabv3+. The proposed approach tries to ensure that the pixel-wise labeling are less depends on the adjacency.

The main contribution of this paper is the introduction of the gradient-based method to the CNN-CRF-based semantic segmentation. The proposed approach works both the levels of directional and scale, while the previous works only focus on using the gradient-based method from scale perspective. The claimed benefits of doing this is to bring better model that helps brining the boundary-caused class weights confusion down.

**Summary Of The Review:**

On the positive side, this paper brings the new gradient-based method perspective into the semantic segmentation, which is interesting. The paper has also shown good results on various popular benchmarks.

However, the paper has not provided sufficient experiments to back up the claims. The improvements of the paper on those benchmarks could be due to the better hyper-parameter search instead of the things claimed in the paper. It is also a bit disappointed to see that the paper does not refer and discuss the difference between the proposed approach with the prior work PCGrad.

---

> ### Author Response · Authors · 2022-11-13
> **Thanks and response to Reviewer qXNf (2)**
>
> **Q3:** How the proposed approach generalize the formulation to the multi-class classification cases?
>
> **A3:** Thanks for your interests. Below is our derivation of generalizing the formulation to multi-class classification cases.
>
> Firstly,  The multi-class cross-entropy loss can be defined as: $L_{k} = -\ln{P_{k}}$ (We simplify the cross-entropy loss as the label is a one-hot form) where $P_{k}$ can be calculated by softmax function :
>
> $P_{k} = softmax(Y_{k}) = \frac{e^{Y_{k}}}{(\sum\limits_{m \neq k}{e^{Y_{m}}})+e^{Y_{k}}}$  and $Y_{k} = W^T * F_{k}$
>
> ($Y_{m}$ is calculated by other class vectors,  and it is unrelated with $W$and $Y_{k}$).
>
> Next, we calculate the gradient $\nabla{W}$ following equation (11): $\nabla{W} ={\frac{\partial L_{k} }{\partial W}=\frac{\partial L_{k}}{\partial P_{k}}\cdot \frac{\partial P_{k} }{\partial {Y_{k}}}\cdot \frac{\partial {Y_{k}}}{\partial W}}$.
>
> Below we provide the derivative of each term repectively.
>
> $\frac{\partial L_{k} }{\partial P_{k}} = - \frac{1}{P_{k}}$,
>
> $\frac{\partial P_{k} }{\partial {Y_{k}}} = \frac{((\sum\limits_{m \neq k}{e^{Y_{m}}})+e^{Y_{k}})\cdot e^{Y_{k}}-(e^{Y_{k}})^2}{((\sum\limits_{m \neq k}{e^{Y_{m}}})+e^{Y_{k}})^{2}}= \frac{ e^{Y_{k}}}{(\sum\limits_{m \neq k}{e^{Y_{m}}})+e^{Y_{k}}} - (\frac{ e^{Y_{k}}}{(\sum\limits_{m \neq k}{e^{Y_{m}}})+e^{Y_{k}}})^2$,
>
> and $\frac{\partial {Y_{k}}}{\partial W} = F_{k}$.
>
> Note that $P_{k} = \frac{e^{Y_{k}}}{(\sum\limits_{m \neq k}{e^{Y_{m}}})+e^{Y_{k}}}$. So we can simplify $\frac{\partial P_{k} }{\partial {Y_{k}}}$ by $\frac{\partial P_{k} }{\partial {Y_{k}}} = P_{k} - P_{k}^2 = P_{k} \cdot (1-P_{k})$
>
> As a result, $\nabla{W}$can be formulated as : $\nabla{W} = - \frac{1}{P_{k}} \cdot P_{k}  \cdot (1-P_{k}) \cdot F_{k} = - (1-P_{k}) \cdot F_{k}$
>
> In this way, the class weights in the next iteration will be updated:
> $W^{*} = W - \nabla{W} = W + (1-P_{k}) \cdot F_{k}$.
>
> The derivation is finished and we can find that it is the same as the equation (12) in the two-class case in our paper. We hope that it will eliminate your concern.

---

> ### Author Response · Authors · 2022-11-13
> **Thanks and response to Reviewer qXNf (1)**
>
> **Q1:** the main weakness is that the paper does not really refer the good literature, making it appears that the gradient-based method is introduced the first time and the authors apply that to the semantic segmentation. In particularly, I would consider the proposed approach is very similar to the other works such as PCGrad.
>
> **A1:** Thanks for your advice. We are willing to cite PCGrad and add the discussion of the difference between PCGrad and E-CRF in our final paper. Below is the discussion.
>
> **Difference between Projecting Conflicting Gradients (PCGrad) and E-CRF:**  PCGrad is designed to mitigate a key optimization issue in multi-task learning caused by conflicting gradients, where gradients for different tasks point away from one another as measured by a negative inner product.  If two gradients are conflicting, PCGrad alters the gradients by projecting each onto the normal plane of the other, preventing the interfering components of the gradient from being applied to the network.  The idea behind PCGrad is simple and the method is effective.  PCGrad is a task-level gradient optimization method, mainly focusing on conflicting gradients caused by multiple tasks during training (e.g., in semantic segmentation and depth estimation). E-CRF is a finer-grained pixel-level gradient optimization method. E-CRF mainly aims at mitigating the boundary-caused class weights confusion  in semantic segmentation via adjusting class weights from both scale and direction.
>
> **Q2:** Another issue is that the proposed approach does not provide sufficient experiments to support the claim. I would like to see the fine-grained experiments to show that the proposed approach get real improvements on the classes that are sharing the most adjacent pixels.
>
> **A2:** We present the class weights similarity on Cityscapes (left) and corresponding classwise mIoU improvement (right) between DeeplabV3+ and our E-CRF with ResNet101 as backbone in Figure here (https://anonymous.4open.science/r/E-CRF-8813/cap.JPG).
> It can be seen that the more the class weights similarity drops, the more the improvement is obtained. We also present detailed comparisons in the Table below. The class order is decided by the number of adjacent boundary pixels (descending order).
>
> | Class id | 7 | 11 | 8 | 1 | 3 | 4 | 2 | 6 | 17 | 13 | 9 | 12 | 15 | 5 | 10 | 16 | 0 | 14 | 18 | mean |
> | --- | --- | --- | --- | --- | --- | --- | --- | --- | --- | --- | --- | --- | --- | --- | --- | --- | --- | --- | --- | --- |
> | DeeplabV3+ | 81.12 | 83.79 | 92.32 | 85.94 | 63.18 | 63.80 | 92.59 | 74.25 | 70.71 | 95.29 | 63.78 | 67.37 | 91.20 | 69.62 | 94.55 | 84.59 | 97.74 | 84.60 | 79.76 | 80.85 |
> | E-CRF | 82.23 | 85.05 | 93.56 | 87.34 | 65.36 | 65.62 | 93.99 | 74.98 | 71.60 | 95.40 | 64.44 | 67.35 | 91.43 | 69.71 | 95.42 | 85.45 | 98.74 | 85.46 | 80.49 | 81.77 |
> | Gain | 1.11 | 1.26 | 1.24 | 1.40 | 2.18 | 1.82 | 1.40 | 0.73 | 0.89 | 0.11 | 0.66 | -0.02 | 0.23 | 0.09 | 0.87 | 0.86 | 1.00 | 0.86 | 0.73 | 0.92 |

---

> ### Author Response · Authors · 2022-12-03
> **Looking forward to more discussions !**
>
> Dear Reviewer qXNf:
>
> We sincerely appreciate your time and efforts in reviewing our paper, which would help us improve our final paper!
>
> As the deadline for discussion is approaching, please don’t hesitate to let us know if there are any additional clarifications we can offer.
>
> Look forward to your post-rebuttal rating!

---

> > ### Comment · Reviewer_qXNf · 2022-12-05
> > **Good paper with minor issues**
> >
> > Thanks for addressing my comments and adding more details.
> >
> > I had the concern about this paper since the similarity between the PCGrad and the proposed approach is not mentioned at all. I feel PCgrad is more general approach, while this proposed approach addressed the particular problem in semantic segmentation. However, it does not mean that the proposed approach can just skip the literature.
> >
> > I would like to thank the authors' contribution in making the efforts to show that the proposed approach improves over the previous SOTA approach Deeplabv3.
> >
> > I think it would be better to show how to generalize the proposed approach in the multi-class cross-entropy loss given that is being used in the experiments. I am wondering what's the technique consideration about the trade-off between the multi-class setting and the multiple binary class. It would be interesting to see some of the discussion about this.
> >
> > Overall, I think the paper has its value in the community and worth considering to be accepted. I will keep my original score.

---

> > > ### Author Response · Authors · 2022-12-06
> > > **Thanks and reply to Reviewer qXNf**
> > >
> > > We sincerely appreciate the reviewer for the great encouragement and nice suggestions. Below is our reply, hoping it would remove your concerns.
> > >
> > > **Q1:** I had the concern about this paper since the similarity between the PCGrad and the proposed approach is not mentioned at all. I feel PCgrad is more general approach, while this proposed approach addressed the particular problem in semantic segmentation. However, it does not mean that the proposed approach can just skip the literature.
> > >
> > > **A1:** We are sorry for this mistake that we miss discussing the similarity between PCGrad and our method i.e., E-CRF. This mistake could be made by accident. PCGrad and E-CRF are both gradient-based methods that focus on adjusting the gradient properly to optimize the learning process more effectively and efficiently. By doing so, PCGrad can solve the gradient conflicts in multi-task learning and E-CRF can mitigate the class weights confusion in semantic segmentation.
> > >
> > > Considering that we have discussed the difference between PCgrad and our method below and add it in our updated paper (please see Sec 5),  we will adjust our paper to add the discussion about similarity in our final paper as well.
> > >
> > > Thank the reviewer for this nice suggestion. It will make our paper more complete.
> > >
> > > **Q2:** I think it would be better to show how to generalize the proposed approach in the multi-class cross-entropy loss given that is being used in the experiments. I am wondering what's the technique consideration about the trade-off between the multi-class setting and the multiple binary class. It would be interesting to see some of the discussion about this.
> > >
> > > **A2:** Firstly, we thank the reviewer for this nice suggestion. And we feel very sorry for making such a misunderstanding about the multi-class setting and the multiple binary class.
> > >
> > > In fact, we indeed use multi-class cross-entropy loss in our response before (in "Thanks and response to Reviewer qXNf (2)" part) to deliver our derivation. We speculate that this misunderstanding could be caused by our simplified multi-class cross-entropy loss $L_{k} = -\ln{P_{k}}$.
> > >
> > > We will give a detailed derivation to illustrate how this loss is derivated.
> > >
> > > Generally, the multi-class cross-entropy loss for n-class is calculated by the following formula:
> > >
> > > $L = $$CE$\_$Loss(P,Y) = -\sum_{i=0,i<n} y_{i}\ln{P{i}}$
> > >
> > > $P\in R^{n}$ is the probability vector where $P_{i}$ is the probability of class $i$ and $Y \in R^{n}$ is the one-hot form label where the weight of the right class is $1$ while that of others is $0$.  So we can simplify the $CE$_$Loss(P,Y)$.
> > >
> > > $L = -(0*\ln{P_{0}}+ 0 * \ln{P_{1}} + 0 * \ln{P_{2}} + .... + 1 * \ln{P_{k}} + ....) = -(1 * \ln{P_{k}}) = -\ln{P_{k}}$
> > >
> > > where $P_{k}$ is the probability of the right class $k$.  Thus, we can obtain $L = -\ln{P_{k}}$.  And we just simply replace the symbol $L$ with $L_{k}$ for consistence with $P_{k}$.
> > >
> > > Our derivation is finished, hoping that it will eliminate your concern. We will also follow your suggestion to show how to generalize the proposed approach in the multi-class cross-entropy loss in our final paper.

---

### Official Review · Reviewer_JKCJ · 2022-10-25

**Confidence:** 4
**Clarity, Quality, Novelty And Reproducibility:** 1. Most content of this paper is clea…
**Correctness:** 3
**Technical Novelty And Significance:** 2
**Empirical Novelty And Significance:** 2
**Recommendation:** 6

**Strength And Weaknesses:**

Strength:
1. Although it is well known that boundary pixels are difficult to classify, this paper views this problem from a new perspective – class weight. They identify this as BCWC problem and make statistics to prove that there is a positive relationship between class weight similarity and adjacent pixel number.
2. The authors use both theoretical derivation and figure to prove that the proposed E-CRF can adjust both the scale and direction of optimization, which makes the presentation clearer and convincing.
3. The ablation study is good, fully studying the effects of various components like E-CRF and superpixel.

Weaknesses:
1. It is strange to use superpixel in this paper, which has limited relation to the BCWC problem (the motivation). The superpixel is more like an auxiliary module to boost the performance, since E-CRF alone is not enough for SOTA performance.
2. In the ablation study 4.2.1, there is no obvious advantages of E-CRF over joint-CRF, and the F-score gain is less than 0.3%, and mIoU gain is around 0.5%. Moreover, without superpixel. E-CRF is even worse than joint-CRF with ResNet101 backbone (45.47% vs 45.61%).

3. In the Table 4, the comparison of different boundary refinement methods might be unfair. The backbone and test augmentation (e.g. multi-scale testing) of various methods are not specified. For example, in the ABL paper, they can achieve 52.40 with Swin-B and multi-scale testing. The Table 6 has the same problem.
4. The section 3.3 can only prove that E-CRF can adjust both the scale and direction of optimization but cannot prove that E-CRF can relief BCWC. There lacks a connection between optimization flexibility and BCWC problem.


**Summary Of The Paper:**

In this paper, the authors observe the Boundary-caused Class Weights Confusion (BCWC) problem in semantic segmentation. To solve this problem, they propose E-CRF to fuse the CRF into CNN networks for more effective optimization. The E-CRF owns two advantages: use CRF to purify the feature representation of boundary pixels; enables optimizing class weights from both scale and direction.  Moreover, they add the superpixel as an auxiliary to boost the performance. Experiment results on popular datasets like ADE20K and Cityscapes are provided.

**Summary Of The Review:**

This paper studied the boundary problem in semantic segmentation from a new perspective and observe the BCWC problem. They propose E-CRF and superpixel to solve the BCWC problem. However, the motivation of superpixel is unclear and the explanation for E-CRF is weak. Even worse, there are some unfair comparisons in the experiment, which makes the effectiveness of proposed method unconvincing. Based on these reasons, I think this paper is below the acceptance threshold.
Update: From the response, the authors have addressed my key concern, and I would like to raise my rating.

---

> ### Author Response · Authors · 2022-11-13
> **Thanks and response to Reviewer JKCJ (2)**
>
>
> **Q4:** The comparison of different boundary refinement methods might be unfair. The backbone and test augmentation (e.g. multi-scale testing) of various methods are not specified. The Table 6 has the same problem.
>
> **A4:** Thanks for pointing out this issue. This mistake is made by accident or due to space limits. We are sorry for this confusion. Below are our comparison details.
> In Table 4, we use DeeplabV3+ with ResNet101 as our baseline method. For Segfix, we use the official code (It uses HRNet as backbone) to boost the performance of DeeplabV3+. For DecoupleSegNet which is constructed based on DeeplabV3+, we also use the official code (It uses ResNet101 as backbone). All the models are trained on ADE20K for 80K iterations with batch size set to 16. When testing, we adopt the single-scale testing strategy (i.e., raw image) because using a single scale (i.e., raw images) when comparing with baselines or performing ablation studies is a traditional default setting in the semantic segmentation field[1,2,3,4,5]. The goal is to eliminate the effect of other elements (e.g., image augmentation).
> In Table 6, all the methods adopt ResNet101 as backbone and leverage multi-scale testing and left-right flipping strategies. We report the scores of compared models from their published papers.
>
> [1]  Encoderdecoder with atrous separable convolution for semantic image segmentation. ECCV2018
>
> [2]  Ccnet: Criss-cross attention for semantic segmentation. ICCV2019
>
> [3]  Gated fully fusion for semantic segmentation. AAAI2020
>
> [4]   Ocnet: Object context network for scene parsing. IJCV2021
>
> [5]   Learning statistical texture for semantic segmentation CVPR2021

---

> ### Author Response · Authors · 2022-11-13
> **Thanks and response to Reviewer JKCJ (1)**
>
> **Q1:** The section 3.3 can only prove that E-CRF can adjust both the scale and direction of optimization but cannot prove that E-CRF can relief BCWC. There lacks a connection between optimization flexibility and BCWC problem.
>
> **A1:** First, we thank the reviewer for admitting one of our contributions that our E-CRF extends optimization flexibility. To remove your concern, we have to raise your attention that **the optimization process is directly applied to the class weight matrix (in the final 1x1 convolutional classification layer)**.
>
> In E-CRF, we calculate the gradient using equation (14)
> ($- \nabla{W} = (1-P_{k}^{*}) \cdot \frac{1}{Z_{k}}(\sum_{j\in G}w_{j}F_{j}+F_{k})$)
>
> and update the class weight using $W^{*}=W - \nabla{W}$  (we ignore the learning rate for simplicity). **In other words, we can adjust the class weight from both the scale and direction to make the class weights more discriminative to decrease the class weights similarity (or class weights confusion).** The experimental curve in Figure 1(a) also supports our idea. We are sorry for this confusion and will add this discussion in our final paper.
>
> **Q2:** It is strange to use superpixel in this paper, which has limited relation to the BCWC problem (the motivation). Moreover, without superpixel, E-CRF is even worse than joint-CRF with ResNet101 backbone.
>
> **A2:** **It is not strange to use superpixel and it is also closely related to the BCWC problem.** The logic is below. The capability that we can adjust the class weight from both the scale and direction is a double-edged sword as the noise information can also have a direct influence on the class weights (likely to hinder the optimization for the BCWC problem). This is not what we really want. So we introduce superpixel and try to leverage its local prior to suppress the noise. This may also explain why E-CRF without superpixel may be a little inferior to joint-CRF with ResNet101 backbone. We conduct another interesting experiment where superpixel is added into joint-CRF (45.61%) with ResNet101 backbone. We train this model following the same setting in our paper. The final mIoU is 45.72% (0.11% improvement). While using superpixel in E-CRF, **our real E-CRF** achieves 46.02% mIoU (0.55% improvement). This result supports our opinion and indicates that $\psi_{p}^{f}$(pairwise) and $\psi_{s}^{f}$(superpixel) are complementary in our E-CRF. We are sorry for this confusion and will add this discussion in our final paper.
>
> **Q3:** In the ablation study 4.2.1, there is no obvious advantages of E-CRF over joint-CRF, and the F-score gain is less than 0.3%, and mIoU gain is around 0.5%.
>
> **A3:** **We need to explain that the gains achieved by E-CRF are actually non-trivial.**
>
> Why that the F-score gain is less than 0.3% is non-trivial?
>
> Firstly, the ADE20K dataset is more challenging as it contains more categories and the scenes are more complicated, especially, in the boundary. In contrast, our E-CRF outperforms Joint-CRF, Vanilla-CRF, and baseline on Cityscapes dataset by 1.14%, 2.11%, and 3.81% in F-score respectively. Please see Appendix A.3.
>
> Secondly, in this work, we use 1-pixel boundary F-score to measure the boundary segmentation performance. Please see Sec 4.1. This metric is a very fine-grained and precise measurement. It only tolerates the boundary pixels to have 1-pixel-level distance dislocation. Such a setting makes the improvement not obvious. Of course, we can use other slack metrics, for example, 3-pixel level and 5-pixel level to amplify the improvement. However, we avoid this because it may involve measure errors and can not reflect the true performance precisely.
>
> Why that mIoU gain is around 0.5% over joint-CRF is non-trivial?
>
> Firstly, the mIoU gain is obtained over 150 categories in ADE20K. In the semantic segmentation field, the ADE20K dataset is the most difficult dataset due to its complicated scenes and diverse categories.
>
> Secondly, considering other methods published at various top conferences (e.g., CVPR, NeurIPS, ICCV, and ECCV) in Table 6, 0.5% mIoU gain is non-trivial. For example, STLNet (CVPR2021) is 0.21% higher in mIoU than CPNet (CVPR2020). RecoNet (ECCV2020) is 0.16% higher in mIoU than APCNet (ICCV2019).
> Please see Sec 4.4.

---

### Author Response · Authors · 2022-11-13
**General response**

We sincerely appreciate all reviewers’ time and efforts in reviewing our paper. And we also thank all reviewers for their insightful and constructive suggestions, which help a lot in further improving our paper. According to reviewer's suggestions, we have revised our paper. Below are the main modifications.

- add more contents to emphasize that the optimization process is directly applied to the class weight matrix (in the final 1x1 convolutional classification layer)
- clarify the reason of using superpixel more specifically and clearly
- provide more comparison details
- add a new section to discuss the difference between E-CRF and three related works including PCGrad, OCNet, and SegFix.
- supplement the computational costs of Vallina-CRF and Joint-CRF in Table 5.

We hope our pointwise responses below could clarify all reviewers’ confusion and alleviate all concerns. We thank all reviewers’ time again.

---

### Decision · Program_Chairs · 2023-01-20

**Decision:**

Accept: poster

**Justification For Why Not Higher Score:**

The reviewers were positive but not excited about the paper. Three reviewers recommended it "marginally above the acceptance threshold" with high confidence, and one rated it "marginally below the acceptance threshold" with low confidence.

**Justification For Why Not Lower Score:**

The reviewers were positive but not excited about the paper. Three reviewers recommended it "marginally above the acceptance threshold" with high confidence, and one rated it "marginally below the acceptance threshold" with low confidence.

**Metareview: Summary, Strengths And Weaknesses:**

Four experts reviewed the paper. Three reviewers recommended it "marginally above the acceptance threshold" with high confidence, and one rated it "marginally below the acceptance threshold" with low confidence. Hence, the decision is to recommend the paper for acceptance. However, the reviewers did raise some concerns about the experiments and motivation. The authors are encouraged to make the necessary changes to the paper to the best of their ability.

**Note From Pc:**

if the above contains the word "oral" or "spotlight" please see: "oral" presentation means -> notable-top-5% and "spotlight" means -> notable-top-25%. As stated in our emails, we are disassociating presentation type from AC recommendations